# Methicillin-Resistant *Staphylococcus aureus* from Peninsular Malaysian Animal Handlers: Molecular Profile, Antimicrobial Resistance, Immune Evasion Cluster and Genotypic Categorization

**DOI:** 10.3390/antibiotics11010103

**Published:** 2022-01-14

**Authors:** Minhian Chai, Muhammad Zikree Sukiman, Amirah Huda Kamarun Baharin, Insyirah Ramlan, Lennard Zhunhoong Lai, Yeewen Liew, Pavitra Malayandy, Noor Muzamil Mohamad, Siewshean Choong, Siti Mariam Zainal Ariffin, Mohd Faizal Ghazali

**Affiliations:** 1School of Animal, Aquatic and Environmental Sciences, Faculty of Bioresources and Food Industry, Universiti Sultan Zainal Abidin, Besut 22200, Terengganu, Malaysia; amthonychai@gmail.com (M.C.); muhammadzikree@yahoo.com (M.Z.S.); amirahhuda849@gmail.com (A.H.K.B.); insyirahramlan48@gmail.com (I.R.); lennard9714@gmail.com (L.Z.L.); yeewenliew@gmail.com (Y.L.); pavitramalayandy@gmail.com (P.M.); 2Centralised Laboratory Management Centre, Universiti Sultan Zainal Abidin, Besut 22200, Terengganu, Malaysia; nmuzamil@unisza.edu.my; 3Department of Clinical Studies, Faculty of Veterinary Medicine, Universiti Malaysia Kelantan, Pengkalan Chepa 16100, Kelantan, Malaysia; shean.cs@umk.edu.my; 4Department of Veterinary Preclinical Sciences, Faculty of Veterinary Medicine, Universiti Putra Malaysia, Serdang 43400, Selangor, Malaysia; sitimariam_za@upm.edu.my

**Keywords:** antimicrobial resistance, antibiotic resistance genes, virulence genes, *spa* typing

## Abstract

*Staphylococcus aureus* (*S. aureus*) infections, particularly methicillin-resistant *Staphylococcus aureus* (MRSA) in humans and animals, have become a significant concern globally. The present study aimed to determine the prevalence and antibiogram of *S. aureus* isolated from animal handlers in Peninsular Malaysia. Furthermore, the genotypic characteristics of *S. aureus* isolates were also investigated. Nasal and oral swab samples were collected from 423 animal handlers in Peninsular Malaysia. The antibiogram profiles of *S. aureus* against 18 antibiotics were established using a Kirby–Bauer test. The genotypic profile of *S. aureus*, including the presence of antimicrobial resistance (AMR), virulence genes and *spa* genotypes, was investigated using molecular techniques. The overall carriage rate of *S. aureus*, MRSA and MDRSA was 30.5%, 1.2% and 19.4%, respectively. *S. aureus* was highly resistant against penicillin (72.3%) and amoxicillin (52.3%). Meanwhile, gentamicin and linezolid were fully effective against all the isolated *S. aureus* from animal handlers. It was observed that animal handlers with close exposure to poultry were more likely to carry *S. aureus* that is resistant to tetracycline and erythromycin. *S. aureus* isolates harboured tetracycline resistance (*tet*K, *tet*L and *tet*M), erythromycin resistance (*ermA*, *ermB*, *ermC* and *msrA*) and immune evasion cluster (IEC) genes (*scn*, *chp*, *sak, sea* and *sep*). Seventeen different *spa* types were detected among the 30 isolates of MDRSA, with t189 (16.7%) and t4171 (16.7%) being the predominant *spa* type, suggesting wide genetic diversity of the MDRSA isolates. The present study demonstrated the prevalence of *S. aureus* strains, including MRSA and MDRSA with various antimicrobial resistance and genetic profiles from animal handlers in Peninsular Malaysia.

## 1. Introduction

*Staphylococcus aureus (S. aureus)* is an opportunistic, zoonotic pathogen that is capable of causing a wide array of infections, ranging from mild skin infections to life-threatening bacteremia in both humans and animals [1,2]. Recently, the emergence of antimicrobial resistance (AMR) in common pathogenic bacteria such as MRSA is considered to be a significant health threat, leading to an increase in healthcare costs, treatment failure and deaths [3]. The issue of AMR is more pressing in low and middle-income countries where the data regarding the burden and epidemiology of AMR bacteria in the community and animals are relatively scarce. *S. aureus*, especially MRSA infections, has always been a significant problem in both medical settings and the community, due to its ability to show multiple resistance to various antibiotics, especially towards beta-lactam antibiotics [4]. However, with the recent emergence of multidrug-resistant *S. aureus* strains, such as livestock-associated MRSA (LA-MRSA) in animals, animal handlers such as veterinarians and farmers are equally at risk of being colonized and infected by AMR strains originating from animals [5,6,7]. Previous studies have shown that the prevalence of *S. aureus* among pig farmers ranged from 26.1% to 57.0% [8,9]. Meanwhile, the carriage of MRSA among pig farmers ranged from 2.5% to 6.7% [8,9]. Nonetheless, Oppliger and co-authors reported a higher carriage rate of *S. aureus* and MRSA among farmers and veterinarians as compared to individuals without contact with pigs, indicating that constant exposure to animals may be a risk for *S. aureus* colonization [8]. In addition, Wardy et al. (2015) reported that people with current swine exposure were significantly more likely to carry *S. aureus*, tetracycline-resistant *S. aureus* (TRSA), multidrug-resistant *S. aureus* (MDRSA) and livestock-associated *S. aureus* (LA-SA) than individuals that lacked exposure [9]. It was found that the transmission of MRSA from animals to humans could occur via three routes: direct contact with livestock and pet animals and environmental contamination [4]. Moreover, multiple studies have shown that the handling or consumption of animal products such as raw milk, meat and cheese products can cause the transmission of *S. aureus* and MRSA from animal origin to the human community [2,10,11].

In the last d-ecade, there have been reports of the increasing epidemiological risk of *S. aureus*, particularly from strains that have acquired antimicrobial resistance (AMR) genes [4,12,13]. Previous studies reported that *S. aureus* had acquired resistance to multiple antibiotics (multidrug resistance) through the acquisition of AMR genes that respond to different groups of antibiotics [14]. This is true, especially in the case of methicillin-resistant *S. aureus* when the *mecA* gene was found to be responsible for the resistant trait against beta-lactam antibiotics, including methicillin, penicillin, oxacillin and cefoxitin [15,16]. In addition, methicillin-susceptible *S. aureus* (MSSA) was frequently reported to be harboring AMR genes from other groups of antibiotics simultaneously, including tetracyclines and macrolides [15,17,18]. In addition, *S. aureus* is also capable of producing multiple virulence factors that increase the pathogenicity of the bacteria [4]. These virulence factors include proteins that evade the human host immune responses and promote the colonization of the bacteria to the host cells, enzymes that induce cytolytic effects, as well toxins that induce inflammatory effects and toxic shock syndrome [12,15,19]. Five of these factors include staphylococcal complement inhibitory protein (SCIN), staphylokinase (SAK), chemotaxis inhibitory protein (CHIPS), staphylococcal enterotoxin type A (SEA) and staphylococcal enterotoxin type P (SEP). These five immune evasion modulators are encoded by a group of genes called immune evasion cluster (IEC), consisting of *scn*, *chp*, *sak*, *sea* and *sep*. Past studies reported that IEC genes were frequently found in *S. aureus* lineages well adapted to humans, but not in livestock-associated *S. aureus* and LA-MRSA strains [19,20,21]. Another important virulence factor produced by *S. aureus* is Panton–Valentine leukocidin (PVL) encoded by *lukF/lukS-PV* gene, a bi-component pore-forming toxin capable of causing the destruction and lysis of leukocytes [12]. In addition, toxic shock syndrome toxin-1 (TSST-1), which is encoded by the *tst* gene, is one of the most important pyrogenic toxin superantigens (PTSAgs) produced by *S. aureus* that may lead to high fever, diffuse erythematous rash, desquamation of the skin after onset and hypotension [22]. Meanwhile, Staphylococcal exfoliative toxins (ETs) encoded by *eta* and *etb* genes are responsible for skin and soft tissue infections such as Staphylococcal scalded skin syndrome (SSSS) [23]. In Malaysia, most of the *S. aureus* AMR surveillance studies were mainly conducted in patients, medical staff or students with constant exposure to the medical settings [18,24]. Furthermore, the majority of AMR surveillance and genotypic studies tend to focus more on MRSA isolates rather than methicillin-susceptible *S. aureus* (MSSA) isolates [24]. As a result, the previous findings may not necessarily reflect the actual overall AMR status of the *S. aureus* in relatively healthy human populations, particularly in Malaysian animal handlers. Until 2018, the prevalence rate of MRSA among *S. aureus* isolated from human patients and university students ranged from 15.0% to 28.1% [10,14]. In addition, it was also revealed that the clinical MRSA isolates from several hospitals in Malaysia were highly resistant against penicillin (100%), macrolides (>70%) and fluoroquinolone (>59%) antibiotics [14]. Furthermore, data regarding the latest antibiogram and antimicrobial resistance genes profile of *S. aureus* from animal handlers in Malaysia remain scarce. This creates knowledge gaps on the data concerning antimicrobial resistance and the genotypic profile of Malaysian MSSA isolates. Realizing this knowledge gap, the present study was carried out with the aim to determine the carriage rate, phenotypic antimicrobial resistance profile and genotypic characteristics of *S. aureus* isolated from animal handlers in Peninsular Malaysia.

## 2. Results

### 2.1. Carriage Rate of S. aureus and MRSA from Animal Handlers

Among the 846 swab samples, 155, or 18.3% (155/846; 95% CI: 15.8–20.8%), swabs were found harbouring *S. aureus* via polymerase chain reaction (PCR) screening of the nuc gene and considered to be *S. aureus* positive. These 155 isolates originated from 129 individuals; thus, the overall prevalence rate of *S. aureus* is 30.5% (129/423; 95% CI: 26.2–34.8%). Eighty-three *S. aureus* isolates originated from nasal (19.6%, 83/423; 95% CI: 11–28.2%) while 72 isolates originated from oral swab samples (17%, 72/423; 95% CI: 13.5–20.5%). Twenty (4.7%, 20/423; 95% CI: 2.7–6.7%) individuals were carrying *S. aureus* in both the nasal and oral parts. Further PCR assay revealed that six (3.8%, 6/155; 95% CI: 0.9–6.7%) *S. aureus* isolates were *mec*A gene positive, indicating the presence of MRSA among five animal handlers (1.2%; 5/423). Five of the MRSA isolates were from nasal (1.2%, 5/423; 95% CI: 0.2–2.2%) while only one was from oral swab (0.2%, 1/423; 95% CI: 0–0.6%) samples. The *S. aureus* and MRSA carriage rates of veterinarians, pet owners and animal farmers are shown in Table 1. Statistical analysis revealed that there is no significant difference between the carriage rates of *S. aureus* and MRSA in categorical variables (*p* > 0.05).

### 2.2. Antibiotic Susceptibility Profile of S. aureus and MRSA from Animal Handlers

The antibiotic susceptibility testing result of *S. aureus* was summarized in Table 2. In the current study, 80.0% (124/155) of *S. aureus* isolates showed resistance against one or more antibiotics. The findings also revealed that *S. aureus* showed the highest resistance rate towards penicillin (72.3%, 112/155; 95% CI: 65.2–79.4%) and amoxicillin (53.5%, 83/155; 95% CI: 45.7–61.3%). Meanwhile, MRSA isolates showed the highest resistance rate against penicillin (100%; 6/6), cefoxitin (100%; 6/6), amoxicillin (83.3%, 5/6; 95% CI: 53.5–100%) and chloramphenicol (50.0%, 3/6; 95% CI: 10.0–90.0%). Gentamicin and linezolid were fully effective (100% susceptible) against all the isolated *S. aureus* from animal handlers. In addition, no significant differences were observed when comparing the antibiotic resistance profile of the *S. aureus* according to categories of animal handlers (veterinarian, pet owner and animal farmer). However, individuals with constant contact with poultry (7/22; 31.8%) were more likely to carry *S. aureus* that showed resistance against tetracycline as compared to ruminants (8/77; 10.4%) and pet animals (3/20; 15.0%). Similarly, *S. aureus* from animal handlers with constant contact with poultry (11/22; 50%) showed a higher resistance rate against erythromycin as compared to ruminants (4/77; 5.2%) and pet animals (6/17; 35.3%) handlers. Meanwhile, individuals with constant exposure to ruminant animals (34/77; 44.2%) were less likely to carry *S. aureus* resistant to penicillin compared to poultry handlers (19/22; 86.4%), pet owners (18/20; 90.0%) and aquatic animal handlers (6/9; 66.7%). In the present study, 19.4% (30/155; 95% CI: 13.2–25.6%) of *S. aureus* isolates were categorised as multidrug resistant as they showed resistance to antibiotics from three different groups of antibiotics. The MDRSA prevalence rates of veterinarians, pet owners and animal farmers were summarized in Table 1. No significant difference was observed when comparing the carriage rate of MDRSA according to sample groups (veterinarian, pet owner and animal farmer). Multiple antimicrobial resistance index (MARI) assessment (Table 3) revealed that 26 (16.8%; CI: 10.9–22.7%) out of 155 *S. aureus* isolates have an MARI value of 0.2 and above. The relatedness of the *S. aureus* isolates based on their phenotypic antibiotic resistance pattern were displayed in the dendrogram (Figure 1) generated using the unweighted pair group method with arithmetic mean (UPGMA), showing that two major clusters can be formed among the isolates.

### 2.3. Genotypic Characterization of S. aureus

In the current study, 12 (7.7%; 12/155) of the *S. aureus* isolates were found harbouring tet genes, where 3.2% (5/155) of the isolates carried the tetK genes, four (2.6%; 4/155) carried the tetL genes while only one (0.6%; 1/155) isolate was harbouring tetM gene. In addition, the prevalence rate of ermA, ermB, ermC and msrA genes were 1.3% (2/155), 1.3% (2/155), 4.5% (7/155) and 3.2% (5/155), respectively. Both vanA and mecC genes were not detected among the isolated *S. aureus*.

In regard to virulence genes, 44.5% (69/155) of *S. aureus* collected from animal handlers contained at least one of the five IEC genes (scn, chp, sak, sea or sep). Furthermore, three out of six MRSA isolates were found harbouring IEC genes. The scn genes were detected in 53 isolates (36.8%; 53/155), while the occurrence rates of sak and chp genes were 30.3% (47/155) and 8.4% (13/155), respectively. A low number of sea (0.6%; 1/155) and sep (0.6%; 1/155) genes were detected from two different *S. aureus* isolates. No IEC genes were found among the 71 (68 MSSA and 3 MRSA) isolates. Fifty *S. aureus* isolates with the presence of scn were later categorised into IEC types. The predominant IEC type in this study was type H (21 isolates), as shown in Table 4. Meanwhile, the predominant IEC type among MDRSA isolates was type E (5/30) as shown in Appendix A. The luk-PV, tst, eta and etb genes were not detected from the 155 *S. aureus* isolates.

Thirty of the multidrug-resistant *S. aureus* (MDRSA) isolates (5 MRSA and 25 MDR-MSSA) were characterized using spa typing (Figure 2). Seventeen different spa types were identified, with the most prominent spa types identified being t189 (16.6%; 5/31) and t4171 (16.6%; 5/31). Other spa types included t2174 (12.9%; 4/31), t3080 (6.5%; 2/31), t3293 (6.5%; 2/31), t050 (3.2; 1/31), t550 (6.5%; 1/31), t084 (3.2; 1/31), t091 (3.2; 1/31), t127 (3.2; 1/31), t315 (3.2; 1/31), t548 (3.2; 1/31), t605 (3.2; 1/31), t714 (3.2; 1/31), t3937 (3.2; 1/31), t4720 (3.2; 1/31) and t9531 (3.2; 1/31). Among the MRSA group, four different spa types were identified, including t189 (*n* = 2), t3293 (*n* = 2), t4171 (*n* = 1), and t3080 (*n* = 1). Based on Figure 2 it was observed that the MDRSA isolates can be divided into three major clusters where t714, t4171, t4720, t3937 and t3293 belong to cluster 1. The spa types t189, t2174, t315 and t548 were grouped into cluster 2, while t3080, t605, t084, t9531, t127, t091, t050 and t550 were categorized into cluster 3. The genotypic profiles of MRSA and MDR-MSSA from animal handlers in Peninsular Malaysia were shown in Appendix A. Based on the findings, it was observed that genotypic profiles of the MDRSA were different, with the exception of MRSA 3 and MRSA 4.

## 3. Discussion

Since its first discovery, information regarding *S. aureus* strains, especially MRSA, has rapidly changed around the globe [25]. Thus, continuous surveillance of *S. aureus* and MRSA in different communities is highly required to obtain the latest data regarding the prevalence, antibiogram profile and genotypic characteristics of these bacteria. Moreover, the antibiogram profiles obtained in this study can be used to better understand the AMR status of *S. aureus* in a relatively healthy community and as a reference to select the best antibiotics for the successful treatment of *S. aureus* and MRSA infection in Malaysian animal handlers.

The prevalence rate of *S. aureus* among animal handlers reported in this study is within the range of the reported 20% to 30% nasal carriage rate in the United States and European countries [26]. However, there is also a wide range in the nasal *S. aureus* carriage rate among European countries, with the Hungarian participants having a significantly lower *S. aureus* prevalence rate (12.1%) as compared to a 29.4% carriage rate among participants from Sweden [26]. In Malaysia, findings regarding the carriage rate of *S. aureus* among local individuals with constant exposure to animals such as farmers, pet owners and veterinarians need to be updated. Furthermore, the sample sizes of animal handlers in the previous studies were relatively small (less than 100 participants), with more emphasis on MRSA isolates [27,28]. In 2013, Neela and co-authors reported that only three (3.61%; 3/83) *S. aureus* samples were isolated from Malaysia poultry farm workers, with none of the isolates being *mec*A positive [27]. Another study by Akilu and his co-authors (2012) demonstrated a higher MRSA prevalence rate, where 7.1% (2/28) of staff from University Veterinary Hospital of Universiti Putra Malaysia were found to be MRSA positive as compared to the 1.2% recorded in this study [28]. Nonetheless, a study conducted in the Netherlands reported a lower carriage rate, where 0.6% of adults living in close proximity to livestock farms were colonized with MRSA [7]. The reasons for the low MRSA carriage rate presented in this study remain unknown. However, it is agreed that factors such as genetic factors, environmental factors, type of samples or sampling sites (only on one site or multiple sites including the anterior nares, pharynx, skin and perineum), cultivation of bacteria (with or without enrichment) and study design (cohort study with repeated sampling or cross-sectional study with one-time sampling) can influence the results of such studies [26].

In the current study, *S. aureus* from animal handlers showed resistance against 18 different antibiotics, with 19.4% (30/155) of *S. aureus* isolates categorised as multidrug resistant. This finding is similar to the 19.4% MDRSA prevalence rate among individuals with close contact with livestock [9]. Antibiotic susceptibility tests revealed that *S. aureus* from animal handlers was highly resistant to penicillin (72.3%). This finding is lower than the 90% to 100% penicillin resistance rate among *S. aureu*s from hospital and university students in Malaysia [18,29]. Meanwhile, MRSA isolates showed a high resistance rate against beta-lactam antibiotics, particularly penicillin (100%; 6/6), cefoxitin (100%; 6/6) and amoxicillin/clavulanate (83.3%; 5/6). This is not surprising as MRSA strains are often resistant to nearly all of the beta-lactam antibiotics due to the production of PBP2a [2]. Despite that, all of the *S. aureus* isolates were fully sensitive towards linezolid and gentamicin, suggesting that these antibiotics can be used to treat persistent *S. aureus* infections among animal handlers. Nonetheless, as clinical tests were not carried out in the study, the effectiveness of these antibiotics against *S. aureus* infections remains inconclusive. Further MAR index assessment revealed that 16.7% *S. aureus* had an index value of 0.2 and above. MAR index values of 0.2 and above have been used to differentiate low- and high-risk regions where antibiotics are overused, thus suggesting that 16.7% of the *S. aureus* isolates from Malaysian animal handlers were exposed to environments with high antibiotic use and high selective pressure [30]. The main reason for the presence of *S. aureus* strains that exhibit phenotypic resistance traits among local animal handlers remains uncertain. It is well known that repeated therapeutic and indiscriminate usage of antibiotics will cause the development of antimicrobial resistance in microorganisms [31]. In the present study, the majority of participants (81.4%) declared that no antibiotics were used by themselves within a three-month period. Furthermore, no significant differences were observed when analysing the carriage rate of MDRSA or *S. aureus* with AMR traits according to the usage of antibiotics within a three-month period, suggesting short-term antibiotic usage alone was unlikely to cause the emergence of AMR. However, it is possible that these participants were colonized by *S. aureus* with AMR traits through direct contact with infected animals or humans with a prolonged antibiotic usage history, environmental agents and animal food products contaminated with AMR bacteria [10,11,32,33,34]. In the present study, *S. aureus* from individuals with exposure to poultry were found to have a higher resistance rate against tetracycline and erythromycin as compared to other animal species, suggesting there might be a relationship between these two variables. Previous research has indicated that individuals that were frequently exposed to live poultry were more likely to carry livestock-associated *S. aureus* strains, such as LA-MRSA [35]. In Malaysia, antimicrobials such as erythromycin and tetracycline antibiotics were permitted to be used in poultry for both disease treatment and prevention purposes [27]. Hence, there is a possibility that *S. aureus* with erythromycin- and tetracycline-resistant traits may have originated from the poultry. However, as no samples were collected from the animals, it is uncertain that the *S. aureus* from poultry origins may be presented with the similar antibiotic resistance pattern. Thus, it would be beneficial for future studies to collect samples from various sources (human, animals and environments) over a longer period to determine the possible origin of AMR bacteria. In most of the cases, phenotypic antibiotic resistance is encoded by the various antimicrobial resistance genes found within the bacteria. Thus, it is coherent to speculate that *S. aureus* isolates resistant against these antibiotics may harbour antimicrobial resistance genes that encode for defensive mechanisms, including the production of defensive enzymes, reduced permeability, enzymatic inactivation, efflux pumps or ribosomal protection [14]. In the present study, *mecA*, *tetK*, *tetL*, *tetM*, *ermA*, *ermB*, *ermC* and *msrA* genes were detected. The *tet*K and *tet*L genes are responsible for the tetracycline efflux pump system, while *tet*M is involved in ribosomal protection [14]. Meanwhile, *ermA*, *ermB* and *ermC* genes are responsible for the ribosomal binding site modification to protect the *S. aureus* against erythromycin antibiotics, while the *msrA* gene encodes for an ATP-dependent efflux pump [36]. The *S. aureus* isolates in this study displayed a higher prevalence of *ermC* (4.5%; 7/155). This finding is in agreement with a study in South Africa where the *ermC* gene is the most frequently found macrolide–lincosamide resistance gene in pigs [36]. Nonetheless, it is observed that certain isolates did not harbour the antimicrobial resistance genes despite showing phenotypic resistance. It is possible that the phenotypic resistance of these isolates may be mediated by resistance genes or mechanisms that are not included in study [37,38].

Immune evasion virulence factor is one of the many vital factors that enable *S. aureus* to adapt and survive in the human host. IEC has been shown to play an essential role in the disruption or inhibition of the normal function of the human immune system, as well as causing food poisoning [20]. Thus, the presence of IEC genes is believed to facilitate the colonization and invasion of *S. aureus* in human hosts. In addition, the absence of *scn* gene in *S. aureus* is considered to be a marker and strong indicator for livestock-derived strains [9,19]. The screening of IEC genes in *S. aureus* isolates showed that 44.5% (69/155) of isolates collected from animal handlers were found to contain at least one of the IEC genes, with the predominant IEC type in this study being Type H. This finding is different compared to other studies that recorded a high prevalence of IEC genes in clinical *S. aureus* (>75%) isolates, with the predominant IEC type being Type B [20,39]. However, several studies have also demonstrated that isolates from humans with occupational exposure to animals may not harbour the IEC genes, possibly due to the transmission and colonization of *S. aureus* from animals [19,21]. Further analysis of IEC composition among MDRSA (Appendix A) also revealed that more than half (56.7%; 17/30) of the isolates were lacking IEC genes, highlighting the possibility of transmission of MDRSA from animals to animal handlers. Thus, the prevalence of AMR in *S. aureus* from animal sources in Malaysia should be monitored closely. Nonetheless, most of the *S. aureus* samples were generally susceptible to the tested antibiotics and lacked virulence factors, suggesting the isolates were of less pathogenic significance to humans [40].

Molecular typing of the *S. aureus* is important to understand the dissemination and epidemiology of bacteria. In the present study, *spa* typing was used to identify the genotypes and determine the relatedness of the isolated MDRSA. According to a study in 2018, the most prominent *spa* types among clinical *S. aureus* isolates in the Asia region were t307 and t002 [41]. Meanwhile, the most prominent *spa* type in clinical isolates in Malaysia was t307 [41]. The previous findings were different to this study, which demonstrated that the most prominent *spa* types identified were t189 (16.6%; 5/31) and 4171 (16.6%; 5/31). The presence of MRSA and MSSA t189 was previously reported among *S. aureus* isolated from hospital patients, university students and poultry farm workers in Malaysia, suggesting the widespread of this genotype among *S. aureus* in local settings [27,42]. Moreover, *S. aureus* t189 was also reported as the prominent genotype in clinical isolates from the Netherlands, Italy, Korea, China and Taiwan [41]. In addition, the present study is the first to report the presence of MRSA t14171 among Malaysian animal handlers. In the past, *spa* type t4171 was only reported in clinical MSSA isolates and MSSA/MRSA isolates from swine in Malaysia [17,42]. The finding of both MRSA t4171 and MSSA t4171 in animal handlers suggests that these genotypes may be unique and common in various local settings. Other *spa* types, such as t548, t050, t3239, t084, t095, t3080, t9531, t605, t550, and t714, detected in this study were reported for the first time in Malaysia. Even though some of the isolates shared the similar *spa* types, their antimicrobial resistance and virulence gene profiles are different, showing that the MDR trait may not be specific to certain strains or locations. Thus, continuous surveillance of *S. aureus* in various communities and environments should be carried out in order to have a more holistic insight on the distribution and emergence of AMR strains. Future genotyping study of local MDRSA strains such as MRSA t4171 using multilocus sequence typing (MLST) and whole genome sequencing (WGS) techniques should be carried out to provide data for better understanding and comparison with international MSSA/MRSA strains. Furthermore, policies and programs that promote the responsible usage of antimicrobials as well as good bacterial infection prevention and control measures among animal handlers in Peninsular Malaysia should be strengthened to prevent the emergence of AMR in local animals.

## 4. Materials and Methods

### 4.1. Swab Sample Collection, Isolation and Genotypic Identification of Bacteria

The sample size of animal handlers was determined using GraphPad StatMate© software (5% absolute precision and 95% level of confidence) for cross-sectional and random sampling. The expected *S. aureus* carriage rate of 30% was used to determine the maximum sample size. Informed consent was obtained from all recruited volunteers included in the study. In the present study, 846 swab samples (423 nasal and 423 oral swabs) were collected from 423 animal handlers in various states of Malaysia (Terengganu, Kelantan, Pahang, Johor, Selangor, Perak, Penang, Kedah, Negeri Sembilan, Perlis, Malacca and Kuala Lumpur) from the period of September 2019 to December 2019. In the present study, an animal handler was defined as an individual with frequent exposure to domestic livestock and companion animals, such as animal farmers (*n* = 343), pet owners (*n* = 70) and veterinarians (*n* = 10). Nasal swab samples were collected by inserting the swab (approximately 2 cm) into the nostril. The swab was rotated several times against the nasal wall and the procedure was repeated in the other nostril using the same swab. Meanwhile, oral swabs were taken by swabbing the back of the throat, roof of the mouth and cheek for several seconds. Swab samples were carefully placed into the falcon tubes containing transport medium and kept in a cooler box at around 4 °C and sent to the Universiti Sultan Zainal Abidin (UniSZA) Microbiology laboratory for further analysis. The swab samples were then streaked onto mannitol salt agar (Thermo Fisher Scientific, Waltham, MA, USA) and incubated at 37 °C for up to 48 h. Yellow colonies with yellow zones growing on mannitol salt agar were considered to be putative *S. aureus* colonies. The putative *S. aureus* colonies were then sub-cultured on nutrient agar (Thermo Fisher Scientific, Waltham, MA, USA) supplemented with 6.5% sodium chloride (NaCl) and incubated at 37 °C for 24 h. Putative *S. aureus* colonies were then subjected to DNA extraction using the heat lysis method, where the bacterial colonies were suspended in 120 µL distilled water and boiled for 5 min at 95 °C [12]. The extracted DNA was kept at −20 °C prior to analysis. The PCR detection method was used to screen for the presence of *S. aureus* and MRSA. The primers used for the detection of *S. aureus* and MRSA were *nuc* (*S. aureus* species-specific gene) and methicillin-resistant genes (*mecA* and *mecC*), respectively, as shown in Table 5 [43,44]. In this study, 3 µL of DNA templates of bacterial isolates were added to 7 µL PCR mixtures containing primers (0.1 µL forward and 0.1 reverse primer), nuclease-free deionized water (1.8 µL) and 5 µL of MyTaq HS Red Mix (Bioline, London, UK). The PCR protocol was run using the Applied Biosystems Veriti 96 Well Thermal Cycler (Thermo Fisher Scientific, Waltham, MA, USA). Amplified PCR products along with a 100 bp DNA Ladder (Promega, Madison, MI, USA) were loaded into 2.0% (*w*/*v*) agarose gel stained with GelRed Nucleic Acid Gel Stain (Biotium Inc., Fremont, CA, US) and gel electrophoresis was run in 1X tris-borate-EDTA (TBE) running buffer (Bioline, UK) at 80 V for 2 h. The gels were visualized using Fujifilm LAS-4000 gel documentation system (Fujifilm Life Science, Cambridge, MA, US). Bacteria isolates that showed the presence of DNA bands at 278 bp (*nuc* gene) were considered to be *S. aureus*. isolates that displayed that the presence of both *nuc* (278 bp) and *mecA* (533 bp) were positively confirmed as MRSA. Meanwhile, bacterial isolates that carried both *nuc* (278 bp) and *mecC* (138 bp) genes were identified as LA-MRSA. Bacterial isolates with the presence of *nuc* genes only were referred to as methicillin-susceptible *S. aureus* (MSSA). *S. aureus* ATCC 700699 was used as the control strain, and were used as the positive control for *nuc* and *mecA* genes detection.

### 4.2. Antibiotic Susceptibility Test (AST) of S. aureus

The antibiotic susceptibility of *S. aureus* isolates was determined using the Kirby–Bauer test on Mueller–Hinton agar (Oxoid, UK), according to the recommendations given by the Clinical and Laboratory Standards Institute (CLSI). The AST was performed using 18 antibiotics as listed in Table 1. The diameter of the inhibition zones for each isolate was measured and compared to antimicrobial susceptibility breakpoints listed in CLSI guidelines [51]. *S. aureus* ATCC 700699 was used as the control strain. *S. aureus* isolates that showed resistance against three or more categories of antibiotics were classified as multidrug-resistant *S. aureus* (MDRSA) [6].

### 4.3. Genotypic Characterization of S. aureus

All MRSA and MDRSA were further screened using PCR to detect the presence of different antimicrobial resistance genes, including tetracycline-resistant (*tet*K, *tet*L, *tet*M and *tet*O), erythromycin-resistant (*ermA*, *ermB*, *ermC* and *msrA*), and vancomycin-resistant genes (*vanA*) as mentioned in Table 4 [45,46]. The prevalence of nine different virulence genes (Table 4), including the immune evasion cluster (*scn*, *chp*, *sak*, *sea* and *sep*), exfoliative toxins (*eta*, *etb*), PVL toxin (*pvl*) and TSST-1 (*tst*) were also investigated [20,48,49,50]. In regard to IEC genes, *S. aureus* isolates were later classified into eight different IEC types according to the combination of genes detected. The presence of the *scn* gene was mandatory for the consideration of the IEC types [20].

### 4.4. Spa Typing of MDRSA

Thirty isolates of non-repetitive MDRSA (both MRSA and MDR-MSSA) were further investigated using *spa* typing. The X region of the *spa* gene of *MDRSA* isolate was amplified using PCR with the primers spa-1095F (5′-AGACGATCCTTCGGTGAGC-3′) and spa-1517R (5′-GCTTTTGCAATGTCATTTACTG-3′) [52]. The PCR amplification started at initial denaturation at 94 °C for 3 min followed by 35 cycles of 94 °C for 30 s, 54 °C for 1 min and 72 °C for 30 s, with the final extension at 72 °C for 7 min. The amplified PCR products were purified and sent to Bio Basic research laboratory (Singapore) for DNA sequencing. The sequences were analysed using BioNumerics 8.0 (Applied Maths, Austin, TX, US) to assign the *spa* type of *S. aureus* isolates. The DNA sequence data of *S. aureus* isolate with unknown *spa* repeat and type were uploaded to Ridom Spa Server website (http://spaserver.ridom.de, last accessed at 6 November 2020) for the assignment of new *spa* type.

### 4.5. Statistical Analysis

The occurrence rate of the *S. aureus* and MRSA as well as different antimicrobial and virulence genes was counted and presented in percentages (%). The 95% confidence interval (CI) of the occurrence rates of *S. aureus*, MRSA and MDRSA among the animal handlers was calculated. Categorical data were analysed using Chi-square or Fisher’s exact test (Minitab 19, 2019), with the 95% confidence interval (*p* < 0.05) being set to indicate the significant difference. The antibiotic resistance percentage was calculated as the proportion of isolates tested that had an inhibition zone below the respective antibiotic breakpoint. The relationships between the antibiotic exposure and overall antibiotic resistance of the isolated *S. aureus* were assessed via the multiple antimicrobial resistance index (MARI). The MARI was calculated as the proportion of antibiotics to which *S. aureus* isolate was phenotypically resistant.

## 5. Conclusions

The present study revealed the carriage rate of *S. aureus* (30.5%) and MRSA (1.2%) of animal handlers from Peninsular Malaysia. The findings also showed that *S. aureus* isolates were highly resistant against penicillin and amoxicillin while showing full susceptibility against gentamicin and linezolid. Subsequently, 19.4% of *S. aureus* isolates were categorized as MDRSA, demonstrating the low prevalence of *S. aureus* with multiple drug-resistant traits among Malaysian animal handlers. Nonetheless, different numbers of antimicrobial resistance (*mecA*, *tetK*, *tetM*, *tetL*, *ermA*, *ermB*, *ermC* and *msrA*) and IEC (*scn*, *sak*, *chp*, *sea* and *sep*) genes were detected among the *S. aureus* isolates, raising concern regarding the further emergence of AMR bacteria among animal handlers through the dissemination of antibiotic-resistant genes. Furthermore, molecular typing showed the presence of seventeen different *spa* genotypes were detected among the MDRSA, with t189 (16.7%) and t2174 (16.7%) been the prominent genotypes. The findings from this study suggest the emergence of MDRSA with great genetic diversity and antimicrobial resistance profiles among animal handlers from different locations of Peninsular Malaysia.

## Figures and Tables

**Figure 1 antibiotics-11-00103-f001:**
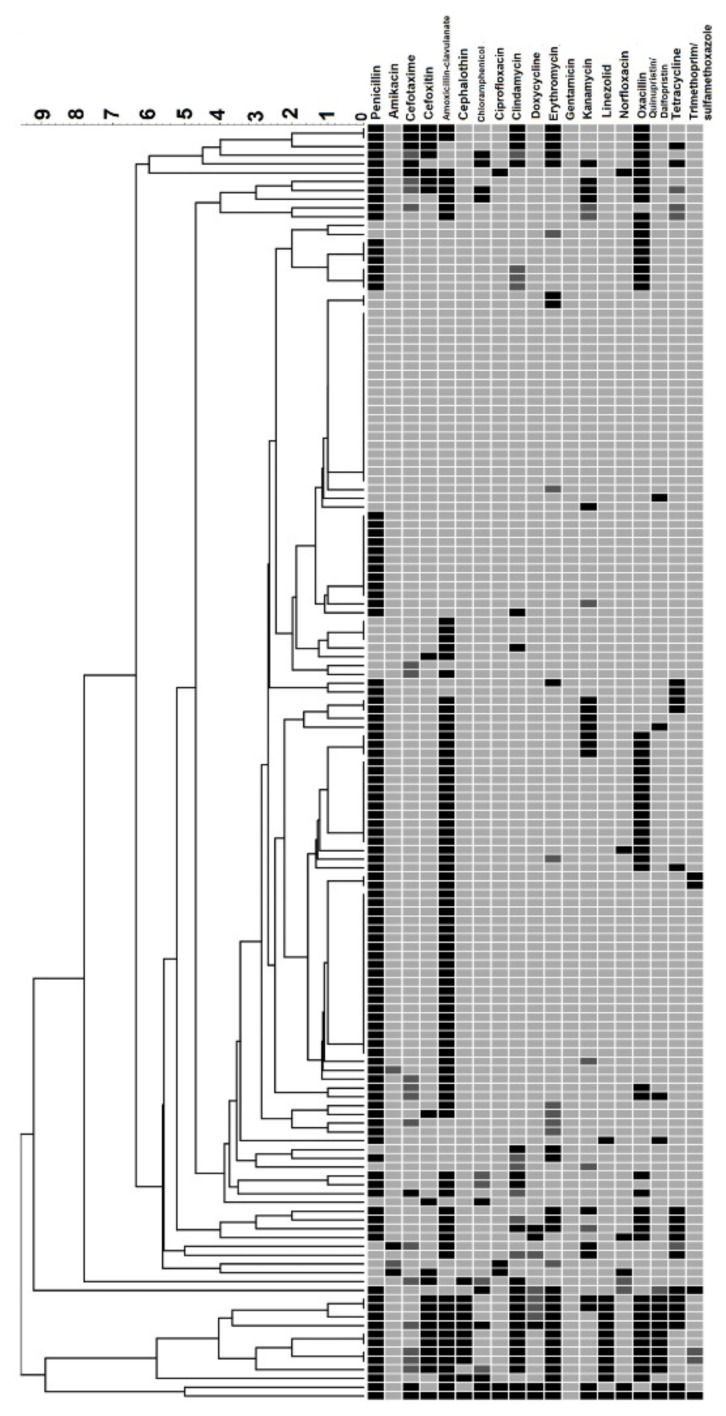
Dendrogram illustrating the relatedness of *S. aureus* based on phenotypic antibiotic resistance pattern. Black = resistant; dark grey = intermediate and light grey = susceptible.

**Figure 2 antibiotics-11-00103-f002:**
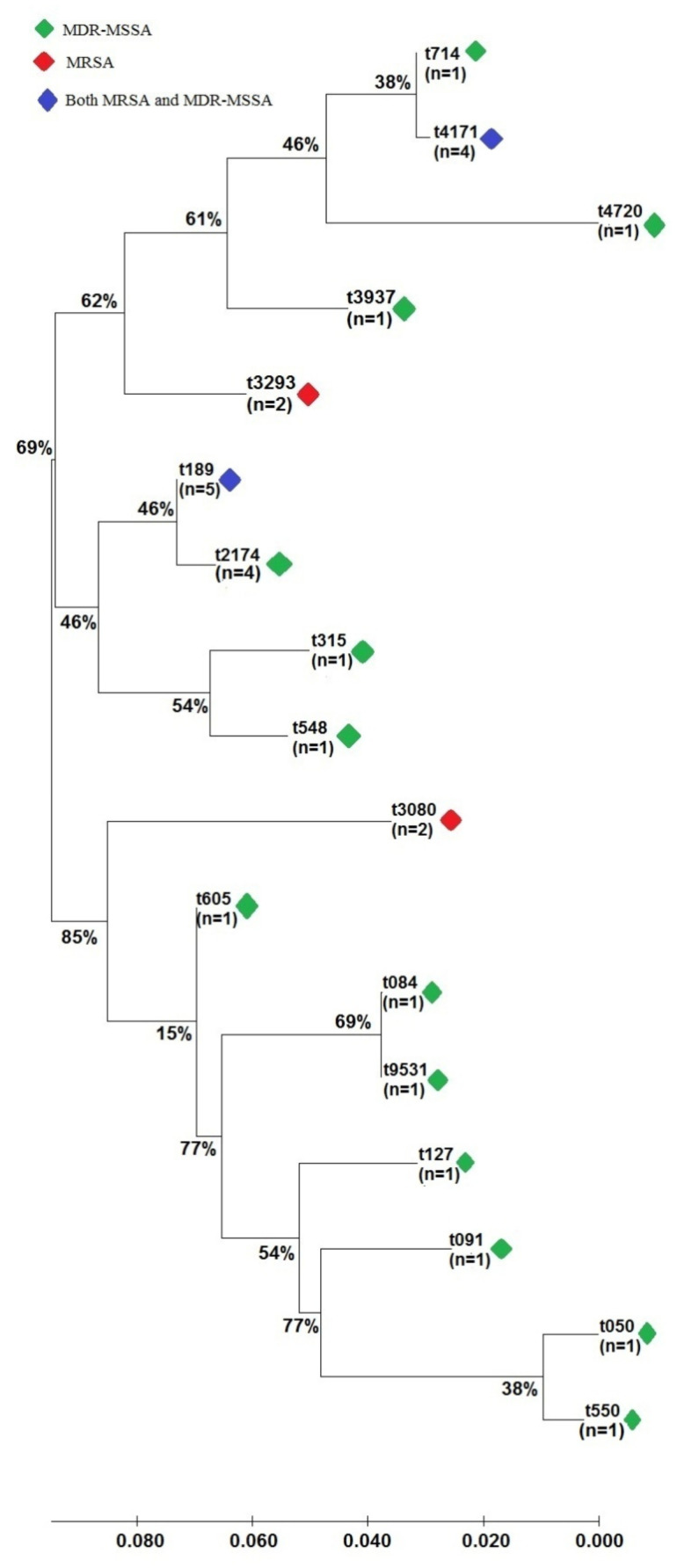
The phylogenetic tree of the spa types of MDRSA isolates was constructed using MEGA version 10. The evolutionary history was inferred using the neighbor-joining method. The optimal tree with the sum of the branch length = 0.45581958 is shown. The tree is drawn to scale, with branch lengths in the same units as those of the evolutionary distances used to infer the phylogenetic tree. The evolutionary distances were computed using the maximum composite likelihood method and are in the units of the number of base substitutions per site.

**Table 1 antibiotics-11-00103-t001:** The carriage rates of *S. aureus*, MRSA and MDRSA among animal handlers (*n* = 423) according to sample groups (veterinarians, pet owners and animal farmers).

No.	Sample Groups	*S. aureus* Positive (%; 95% CI)	MRSA Positive (%; 95% CI)	MDRSA Positive (%; 95% CI)
1.	Veterinarians (*n* = 10)	3 (30.0; 1.0–58.4%)	0 (0; 0%)	2 (20.0; 0–44.8%)
2.	Pet owners (*n* = 70)	20 (28.6; 18.0–39.2%)	1 (1.4; 0–4.2%)	7 (10.0; 3.0–17.0%)
3.	Animal farmers (*n* = 343)	105 (30.6; 25.7–35.5%)	4 (1.2; 0–2.4%)	21 (6.1; 3.6–8.6%)
	Total (*n* = 423)	129 (30.5; 26.2–34.8%)	5 (1.2; 0.2–2.2%)	30 (7.1; 4.7–9.5%)

**Table 2 antibiotics-11-00103-t002:** Antibiogram of *S. aureus* (*n* = 155) isolated from animal handlers in Peninsular Malaysia.

Antimicrobials	Number of Isolates (%)		
	Resistant	Intermediate	Susceptible
Penicillin	112 (72.3)	0 (0)	43 (27.7)
Amoxicillin	83 (53.5)	0 (0)	72 (46.5)
Erythromycin	22 (14.2)	8 (5.2)	125 (80.6)
Clindamycin	19 (12.3)	14 (9.0)	122 (78.7)
Cefoxitin	18 (11.6)	0 (0)	137 (88.4)
Tetracycline	18 (11.6)	4 (2.6)	133 (85.8)
Quinupristin/Dalfopristin	12 (7.7)	1 (0.6)	142 (91.6)
Chloramphenicol	10 (6.5)	4 (2.6)	141 (90.9)
Cephalothin	8 (5.2)	0 (0)	147 (94.8)
Cefotaxime	7 (4.5)	16 (10.3)	132 (85.2)
Norfloxacin	6 (3.9)	2 (1.3)	147 (94.8)
Doxycycline	5 (3.2)	4 (2.6)	146 (94.2)
Ciprofloxacin	5 (3.2)	0 (0)	150 (96.8)
Trimethoprim/sulfamethoxazole	4 (2.6)	2 (1.3)	149 (96.1)
Amikacin	2 (1.3)	0 (0)	153 (98.7)
Gentamicin	0 (0)	0 (0)	155 (100)
Linezolid	0 (0)	0 (0)	155 (100)

Resistant = R; Intermediate = I; Susceptible = S; Penicillin: R ≤ 28 mm, S ≥ 29 mm; Amoxicillin: R ≤ 19 mm, S ≥ 20 mm; Erythromycin: R ≤ 13 mm, I = 14–22 mm, S ≥ 23 mm; Clindamycin: R ≤ 14 mm, I = 15–20 mm, S ≥ 21 mm; Cefoxitin: R ≤ 21 mm, S ≥ 22 mm; Tetracycline: R ≤ 14 mm, I = 15–18 mm, S ≥ 19 mm; Quinupristin/Dalfopristin: R ≤ 15 mm, I = 16–18 mm, S ≥ 19 mm; Chloramphenicol: R ≤ 12 mm, I = 13–17 mm, S ≥ 18 mm; Cephalothin: R ≤ 14 mm, I = 15–17 mm, S ≥ 18 mm; Cefotaxime: R ≤ 14 mm, I = 15–22 mm, S ≥ 23 mm; Norfloxacin: R ≤ 12 mm, I = 13–16 mm, S ≥ 17 mm; Doxycycline: R ≤ 12 mm, I = 13–15 mm, S ≥ 16 mm; Ciprofloxacin: R ≤ 15 mm, I = 16–20 mm, S ≥ 21 mm; Trimethoprim/Sulfamethoxazole: R ≤ 10 mm, I = 11–15 mm, S ≥ 16 mm; Amikacin: R ≤ 14 mm, I = 15–16 mm, S ≥ 17 mm; Gentamicin: R ≤ 12 mm, I = 13–14 mm, S ≥ 15 mm; Linezolid: R ≤ 20 mm, S ≥ 21 mm.

**Table 3 antibiotics-11-00103-t003:** MARI assessment of *S. aureus* (*n* = 155) isolated from animal handlers.

Number of Antibiotic	MARI Value	Number of Isolates	Total (%)
0	0	31	20.0
1	0.06	33	21.3
2	0.11	51	32.9
3	0.16	14	9.0
4	0.22	9	5.8
5	0.27	6	3.8
6 and above	0.33	11	7.1

**Table 4 antibiotics-11-00103-t004:** Summary of IEC types of *S. aureus* isolates (*n* = 155).

IEC Type	IEC Genes Composition	Number of Isolates (%)		Total Number of Isolates (%)
		MSSA (*n* = 149)	MRSA (*n* = 6)	
A	*scn*, *chp*, *sak*, *sea*	1 (6.7)	0 (0)	1 (0.6)
B	*scn*, *chp*, *sak*	10 (6.7)	0 (0)	10 (6.5)
C	*scn*, *chp*	0 (0)	0 (0)	0 (0)
D	*scn*, *sak*, *sea*	0 (0)	0 (0)	0 (0)
E	*scn*, *sak*	17 (11.4)	2 (33.3)	19 (12.3)
F	*scn*, *chp*, *sak*, *sep*	0 (0)	0 (0)	0 (0)
G	*scn*, *sak*, *sep*	1 (6.7)	0 (0)	1 (0.6)
H	*Scn*	21 (14.1)	1 ((16.7)	21 (13.5)
Non-typable	Absent of *scn* gene	17 (11.4)	0 (0)	17 (11.0)
No Type	Absent of all IEC genes	68 (45.6)	3 (50.0)	71 (45.8)

**Table 5 antibiotics-11-00103-t005:** List of primers used for the detection of AMR and virulence genes of *S. aureus*.

No.	Primer	Primer Sequence (5′-3′)	Product Size (bp)	Annealing Temperature (°C)	References
1.	*nuc*	F-GCGATTGATGGTGATACGGTT	278	55	[43]
		R-AGCCAAGCCTTGACGAACTAAAGC			
2.	*mecA*	F-AAAATCGATGGTAAAGGTTGGC	533	55	[43]
		R-AGTTCTGCAGTACCGGATTTGC			
3.	*mecC*	F-GAAAAAAAGGCTTAGAACGCCTC	138	59	[44]
		R-GAAGATCTTTTCCGTTTTCAGC			
4.	*tetK*	F-TCGATAGGAACAGCAGTA	169	55	[45]
		R-CAGCAGATCCTACTCCTT			
5.	*tetO*	F-AACTTAGGCATTCTGGCTCAC	515	55	[45]
		R-TCCCACTGTTCCATATCGTCA			
6.	*tetM*	F-GTGGACAAAGGTACAACGAG	406	55	[45]
		R-CGGTAAAGTTCGTCACACAC			
7.	*tetL*	F-TCGTTAGCGTGCTGTCATTC	267	55	[45]
		R-GTATCCCACCAATGTAGCCG			
8.	*msrA*	F-GGCACAATAAGAGTGTTTAAAGG	940	50	[46]
		R-AAGTTATATCATGAATAGATTGTCCTGTT			
9.	*ermA*	F-GTTCAAGAACAATCAATACAGAG	421	52	[46]
		R-GGATCAGGAAAAGGACATTTTAC			
10.	*ermB*	F-CCGTTTACGAAATTGGAACAGGTAAAGGGC	359	55	[46]
		R-GAATCGAGACTTGAGTGTGC			
11.	*ermC*	F-GCTAATATTGTTTAAATCGTCAATTCC	572	52	[46]
		R-GGATCAGGAAAAGGACATTTTAC			
12.	*vanA*	F-ATGAATAGAATAAAAGTTGC	1032	62	[46]
		R-TCACCCCTTTAACGCTAATA			
13.	*scn*	F-AGCACAAGCTTGCCAACATCG	258	50	[20]
		R-TTAATATTTACTTTTTAGTGC			
14.	*sak*	F-AAGGCGATGACGCGAGTTAT	223	50	[20]
		R-GCGCTTGGATCTAATTCAAC			
15.	*sea*	F-AGATCATTCGTGGTATAACG	408	50	[20]
		R-TTAACCGAAGGTTCTGTAGA			
16.	*sep*	F-AATCATAACCAACCGAATCA	500	50	[20]
		R-TCATAATGGAAGTGCTATAA			
17.	*chp*	F-GAAAAAGAAATTAGCAACAACAG	410	50	[47]
		R-CATAAGATGATTTAGACTCTCC			
18.	*luk-PV*	F-ATCATTAGGTAAAATGTCTGGACATGATCCA	433	55	[48]
		R-GCATCAAGTGTATTGGATAGCAAAAGC			
19.	*tst*	F-TTATCGTAAGCCCTTTGTTG	398	60	[49]
		R-TAAAGGTAGTTCTATTGGAGTAGG			
20.	*eta*	F-CTAGTGCATTTGTTATTCAA	119	55	[50]
		R-TGCATTGACACCATAGTACT			
21.	*etb*	F-ACGGCTATATACATTCAATT	200	55	[50]
		R-TCCATCGATAATATACCTAA			

## Data Availability

The data presented in this study are available in the article and supplementary material.

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
