# Peer review of "Methicillin-Resistant Staphylococcus aureus from Peninsular Malaysian Animal Handlers: Molecular Profile, Antimicrobial Resistance, Immune Evasion Cluster and Genotypic Categorization"

_antibiotics, 2022, doi:10.3390/antibiotics11010103_

Round 1

Reviewer 1 Report

Comments and Suggestions for Authors

Molecular Detection, Antibiogram and Genotypic Characterization of Staphylococcus aureus from Animal Handlers in Peninsular Malaysia

The paper is quite fine, nevertheless some major points should be addressed before publication.

1-The work adds some new insights and uses mostly clear language. But there still have some issues to address.

2-Rewriting the title will help it better convey the work, especially as the authors have just published a new essay on the same subject:

Chai MH, Sukiman MZ, Najib NM, Mohabbar NA, Azizan NANM, Mohamad NM, Ariffin SMZ, Ghazali MF. Molecular detection and antibiogram of Staphylococcus aureus in rabbits, rabbit handlers, and rabbitry in Terengganu, Malaysia. J Adv Vet Anim Res 2021; 8(3):388–395.

If you search for your title on Google, you'll immediately come across your most recent paper.

Suggestion:

Methicillin-resistant Staphylococcus aureus from Peninsular Malaysian Animal Handlers: Molecular Profile, Antimicrobial resistance, Immune Evasion Cluster  and Genotypic Categorization

3-When the abbreviation is the first time to be used in the text. The whole words are needed. Please check throughout the manuscript and revise accordingly (Example. MRSA and MDRSA in abstract).

4-The abstract should be rewritten to be more specific about its goals and outcomes. It also does not include a statement summarizing the conclusion of the article.

5-I do not agree with the

Nevertheless, information regarding the occurrence rate, antibiotic susceptibility and molecular characteristics of S. aureus among animal handlers in Peninsular Malaysia is limited.

You've just published a fresh article on the same topic:

Chai MH, Sukiman MZ, Najib NM, Mohabbar NA, Azizan NANM, Mohamad NM, Ariffin SMZ, Ghazali MF. Molecular detection and antibiogram of Staphylococcus aureus in rabbits, rabbit handlers, and rabbitry in Terengganu, Malaysia. J Adv Vet Anim Res 2021; 8(3):388–395.

6-Keywords should be words not found in the title to assist indexers in cross-indexing your article. Please rewrite.

7-Line 50. Please add reference.

8-More information concerning the prevalence of S. aureus among animal handlers should be included in the introduction section.

9-More references concerning methicillin-resistant S. aureus (MSSA) in Malaysia in the last paragraph.

9-All S. aureus should be inspected (must be italic)

10-When you begin with a number, please. Words, not numbers, should be used.

11-In table 1, How to calculate “Antibiogram” ? Please add footnote for inhibition zone diameter for Resistant, Intermediate and  Susceptible

12-In table 1, Please remove disk potency column.

13-In academic writing, the techniques used should be described in great detail, leaving no space for interpretation. You need to check the section Materials and Methods.

14-Provide the instrument's specifications. Please go over the whole document and make any necessary edits or additions.

15-Line  288, UniSZA Microbiology laboratory, please write full.

16-Line 293, please add sentence about confirmation of S. aureus by Vitek or biochemical test.

17-mecA, mecC and nuc gene, Please include the sequence of primers in the text, or you may create a table that lists all primers used in the text and their suppliers.

18-Line 326, please start by Thirty one (not number).

19-The resolution of the Figure 3 is too low. I can't really see much of the data. Please replace.

20-Why didn't you include a figure similar to your previous work on the subject of

Dendrogram illustrating the relatedness of S. aureus based on their phenotypic antibiotic resistance pattern.

This might improve the quality of the article.

21-The conclusion Section should be expanded and improved.

22-As a result, the quality of all the article has to be enhanced.

24-There must be a major shift in the article's focus and an increase in the number of molecular results figures.

Author Response

Please see the attachment for author's reply. Thank you.

Reviewer 2 Report

Major comments

               The manuscript is very interesting and covers an important topic related to One Health approach. Additionally, it is well written and its presentation is very good. However, I have some major remarks before perform a detailed evaluation of this manuscript.  The remarks are detailed above:

1 – Authors must include information regarding the presence of Staphylococcus on nasal and oral samples from the same human beings.  The same individual had ever both samples positives?

2 – I suggest to authors divide their results in the three groups sampled: veterinarian, pet owners and farmers. What was the number of samples collected for each group? And how were the results? Was there differences on isolation of MRSA, MDRSA?

3 – Additionally, the animal specie related to each individual is interesting to be pointed. The use of antibiotic in animal production is different, i.e., chicken commonly use more than bovine.

This close contact with distinct species would be very interesting to analyze.  The paper is already interesting but authors can obtain more useful information with their data.

4 – The origin of isolates must be submitted as a Supplementary File, to better understand later the isolates included in spa typing.

5 - Abstract must contain a conclusion

6 – Introduction section

The importance of foods of animal origin on the dissemination of MRSA/MDRSA must be cited.  Introduction must describe the importance of virulence factors and  the genes related

7 - Figures 1 and 2 can be removed

8 -  How was sampling scheme and sample calculated? Was it representative? This information is very important to understand the representativeness of this study.

9 - Were samples obtained from more than one individual that work together? This must be useful to understand the spa typing.

10 – The discussion is very comparative and didn’t discuss possible reasons for antimicrobial resistance in the area.

11 - How the One Health concept can be useful to avoid this problem in Malasya?

Minor

Results

L.80 – Swab samples can not harbour genes. Staphylococcys putative colonies had the DNA extracted

81 – italic for S. aureus

Author Response

Please see the attachment for author's reply. Thank you

Reviewer 3 Report

The work is interesting and well-done, regarding a relevant theme and a particular work population (animal handlers) in a definite geographical setting.

The aim of describing prevalence as well as genotypic and antibiotic resistance-related characteristics seems to be noteworthy

Author Response

(The authors gave the same response as above.)

Round 2

Reviewer 1 Report

Accept in present form

Author Response

Please see attachment for author's reply on comments. Thank you.

Reviewer 2 Report

               The manuscript has improved a lot but some remarks need to be solved before its publication, as listed below:

1 – A editoral/grammar revision must be done, i.e., line 20 – “andantibiogram”, l21 – “S.aureusisolates”

2 – According to supplementary file 1, there was 30 MDRSA, while in Table 1 the value was 26 or 25 (data confuse and not well checked). Please, check all values presented in your manuscript.

3 – Line 113 – It is mentioned figure 1 but content is not related to text

4 – Table 1 title must be improved and self-explanatory

5 – I believe that the higher prevalence of MRSA on persons that are closely contacting poultry must be included on abstract. Additionally, possible reasons for this result must be better discussed in discussion section.

6 – There are two Tables 1. Change the number and use them correctly on manuscript. Please, fully check it on whole manuscript

7 – Table 5 is not necessary because the information is already presented as supplementary file as required. Use “Supplementary File 1” on manuscript (i.e.  203)

8 – Check the use of italic for S. aureus (i.e., line 462) in manuscript.

Author Response

(The authors gave the same response as above.)
